# Universities' Scientific and Technological Transformation in China: Its Efficiency and Influencing Factors in the Yangtze River Economic Belt

Lin Zou[1,2], Yi-Wen Zhu[3,4]*

**1** Institute of Geography, Heidelberg University, Heidelberg, Germany, **2** Institute of Management, Shanghai University of Engineering Science, Shanghai, China, **3** Center for Modern Chinese City Studies, East China Normal University, Shanghai, China, **4** Institute of Urban Development, East China Normal University, Shanghai, China

* 515909826@qq.com

**Data Availability Statement:** All relevant data are within the paper and its Supporting Information files.

## Abstract

Universities are important sources of knowledge and key members of the regional innovation system. The key problem in Chinese universities is the low efficiency of the scientific and technological (S&T) transformation, which limits the promotion of regional innovation and economic development. This article proposes the three-stage efficiency analytical framework, which regards it as a complex and interactive process. Avoiding the problem of considering the input and output of university S&T transformation as a "black box" and neglecting the links among different transformation stages. The super efficiency network SBM model is applied to the heterogeneous region of the Yangtze River Economic Belt. Empirical research proves that university S&T transformation has not been effectively improved and the scientific resources invested in universities have not been efficiently utilized in recent years. Generally, Despite the correlation between regional economy and transformation efficiency, the exclusive increase in resources is not enough. Regional openness and the quality of research talents are key factors for the application of technological innovation and technology marketization. Universities should not only pursue the number of research outputs but pay more attention to high-quality knowledge production to overcome difficulties in research achievements transformation.

## 1 Introduction

Universities are important actors in the national or regional innovation system. In recent years, the experience of deep integration of the global economy and science and technology (S&T) has shown that the efficiency of S&T transformation is important to the national or regional economic development. The "Silicon Valley Miracle" and the "Cambridge Phenomenon," among others, have proved that the effective transformation of S&T achievements in universities is extremely relevant to economic growth [1, 2].

**Funding:** This work was supported by National Natural Science Foundation of China (42101175) and China Scholarship Council Postdoctoral Foundation (Grant: 202008310025).

**Competing interests:** The authors have declared that no competing interests exist.

In the era of the knowledge economy, the development of the industrial economy and technological innovation need to effectively transform S&T achievements into market value to gain advantages in competition. Universities are the sources of major S&T achievements and innovative talents. However, in some cases, theory and practice cannot be effectively combined in universities, resulting in many basic scientific achievements with low marketization value. With the rapid growth of China's economy, the government's investment in S&T research in universities has increased. Although Chinese universities produce a large number of scientific research papers or publications every year, the issue is whether these research achievements can be effectively transformed into market value. It has been pointed out by some researchers that the university transformation rate of S&T research achievements in China is still far behind that of western developed countries [3]. Moreover, according to the statistics of the State Intellectual Property Office of China, as of 2017, the effective patent implementation rate of Chinese universities was only 12.9%, quite lower than the national average level of S&T transformation (50.3%) [4, 5]. Thus, long-term and consistent funding or personnel investment in universities are not effective measures. The economic contribution rate of university S&T achievements should be realized by improving the transformation efficiency, rather than blindly expanding S&T research investment. Therefore, the problem of properly evaluating and improving the transformation efficiency of universities' S&T achievements has become crucial to increase the regional innovation level and economic development [6].

The promotion of the development of the Yangtze River Economic Belt is a national strategy that affects the development of China. The Yangtze River Economic Belt has achieved remarkable development in recent years; however, its internal heterogeneity is obvious. The regional innovation levels of S&T resources of universities are quite different. How to realize the sustainable development of innovation resources is a strategic problem that needs to be solved urgently in China.

The transformation of S&T is a complicated process, in which participants in scientific research activities turn knowledge into S&T achievements and, then, into productivity, and realize marketization through the effective utilization of various resources, such as funding and human resources [7, 8]. This paper focuses on the transformation of S&T achievements in universities. Considering universities as the source of knowledge, this paper discusses the efficiency of different processes, from the generation of basic knowledge to the technology marketization. Taking the Yangtze River Economic Belt as a case study, this paper explores (1) the overall efficiency of university S&T transformation in the Yangtze River Economic Belt, and the differences within this heterogeneity region, (2) the features of different S&T transformation stages, and how the efficiency of each stage evolves, and (3) the main factors affecting university S&T achievements transformation in the Yangtze River Economic Belt, as well as the mechanisms to improve university S&T efficiency.

This remainder of this paper is organized as follows. The next section reviews the relevant literature and proposes the analytical framework of S&T transformation efficiency; the third section describes the data and methodology of our research; the fourth and fifth sections discuss the efficiency of S&T transformation and its influencing factors, respectively; finally, in the last section, we offer some concluding remarks.

## 2 Analytical framework

Most innovations come from "reference" rather than "originality" [9], the "silence" and "environmental sensitivity" of S&T make technological innovation and transformation more complex than the simple process of technology introduction [10]; thus, how to efficiently apply and transform the external knowledge is the key to innovation. Innovation participants need

to have the ability to efficiently absorb and apply introduced technologies [11, 12]. Schumpeter (1942) [13] pointed out that promoting innovation by technology introduction could improve innovation efficiency. Universities are important innovation participants, basic knowledge, and technology sources. They participate in complex S&T innovation activities, and organizational openness makes their S&T innovation process different from enterprises. Therefore, the analysis of the transformation efficiency of university S&T activities can help to objectively understand the S&T innovation level and existing problems; also, it plays an important role in optimizing the allocation of S&T resources.

Existing research on university S&T activities and achievements transformation mainly focused on the mode and strategy of S&T achievements transformation, evaluation methods, transformation efficiency, and so on. From the perspective of the transformation model and strategy of university S&T achievements transformation, researches have pointed out that scientific research management ability is one of the most important factors affecting university S&T innovation ability, as improving the efficiency of university management ability is important to improve the efficiency of university S&T achievements transformation. Previous research on the transformation of S&T achievements in American universities showed that there is a negative correlation between the period and efficiency of universities' S&T achievements transformation activities (Saul, 2004). Existing studies on the management of university S&T achievements in China focused on the transformation mechanism [14, 15]. Early research pointed out that the efficiency of university S&T achievements transformation can be improved by increasing the reward for individuals and universities. Later research put forward a new model for university S&T achievements transformation in China, suggesting that the appropriate transformation mode should be selected according to the type of S&T achievements and market demands, and the government and social organizations should cooperate to improve the transformation efficiency of S&T achievements [16]. How to evaluate and improve the transformation efficiency of university S&T achievements has always been a concern; however, the evaluation method of the transformation efficiency is still controversial.

The problems existing in the transformation activities and appropriate suggestions to solve them can be clarified only through scientific evaluation of the transformation efficiency of university S&T achievements. Existing research provided an important basis; however, there are still controversies and problems to be addressed. The university S&T achievements transformation is a complex multi-stage chain process, with multiple inputs and outputs [17]. Most existing efficiency evaluation studies regarded the transformation process as a single input-output stage or focused only on a certain part of the overall activity. However, the analysis of a single stage of the transformation ignores the systematic nature of the process and cannot evaluate efficiency pertinently.

Rothwell (1992) [18] pointed out that technology promotion and market pull are the basis for the formation of a technological innovation chain. Hage (2000) [19] proposed the "idea innovation chain" and divided it into basic research, applied research, and development research. Existing empirical research also divided the technology transformation chain into different stages, which mainly include basic research, applied research, commercialization, and industrialization. The initial stage generally represents the creation of university basic scientific knowledge; the intermediate stage represents the application technology transformation of the innovation chain; and the final stage is the commercialization of technology. Each stage in the whole process of scientific research achievements transformation is an innovation aspect, and different innovation aspects are interrelated to form a complete chain of knowledge innovation and technology application. This study constructs the conceptual model of university S&T achievements transformation as shown below (Fig 1).

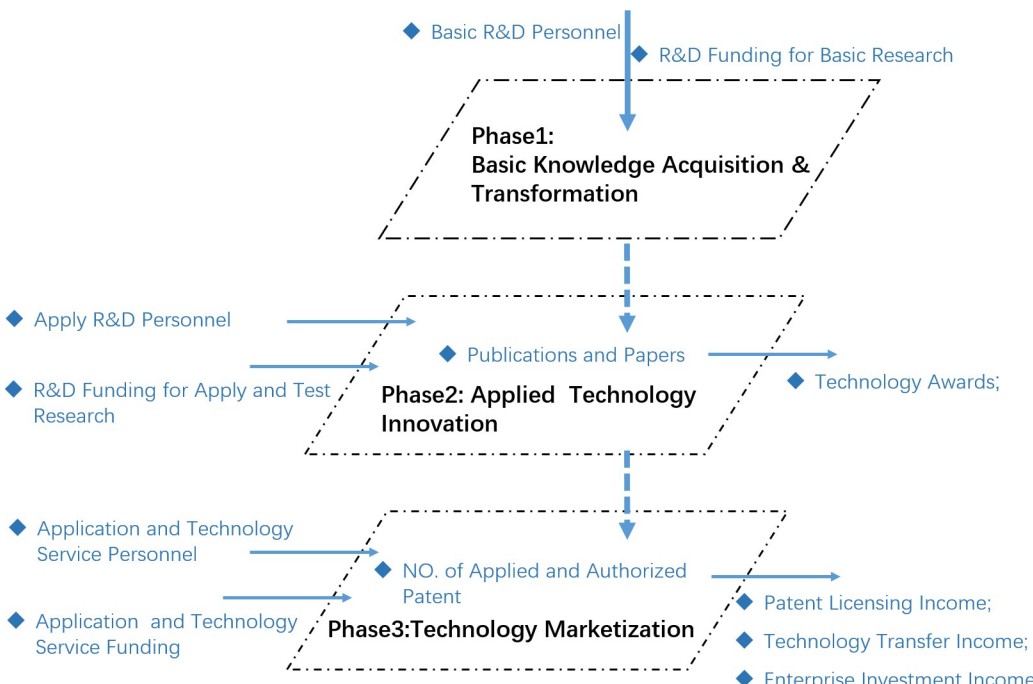

**Fig 1. Conceptual model of University S&T transfer process.**

DEA (Data Envelopment Analysis) [20, 21] and SFA(Stochastic Frontier Analysis) [22–24] have been used to measure the efficiency of S&T achievements transformation in western developed countries and determined the factors restricting the transformation process. DEA is a scientific method for efficiency evaluation, which avoids the inaccuracy of a subjective weight setting. However, in empirical research, the input and output of DEA mainly focused on the initial input resources and final output [25]. The "black box" research method does not include the intermediate stage of transformation into the innovation chain model. With the optimization of efficiency measurement model, based on the innovation process analysis framework, the efficiency research is divided into different stages for evaluation [26]. Focus on the sub stages of innovation process, and establish the relationship among different stages [27]. According to the transformation chain of university S&T achievements, the input-output division is divided into three stages; this method allows us to systematically grasp the total efficiency of the transformation chain and accurately analyze the efficiency problems in different transformation stages.

Universities' S&T transformation processes are complex. The whole transformation is a multi-stage and multi-factor value chain process, including basic knowledge acquisition, technological innovation, and marketization. This paper divides the process into three main stages. The first stage is the acquisition of basic knowledge, and the output of this stage is mainly papers and publications. The second stage concerns the application of technological innovation. Universities take the research results of the first stage as input to jointly carry out application-oriented research and development; the output of this stage is mainly in the form of patents. The third stage is the marketization and value transformation stage of technology. Universities take patents of the second stage as input to jointly carry out experimental research and development; the output of this stage mainly includes patent licensing income, research funds obtained from enterprises, and technology transfer income of universities. In this way, we can effectively distinguish the different university S&T transformation stages, which

include direct investment and indirect investment. The S&T achievements transformation process is not regarded as a black box, but the process and stages of transformation are considered; the input-output correlation between different stages is also taken into consideration [2].

From the perspective of S&T transformation, we consider not only whether higher economic or higher resources input will lead to better efficiency but also how the S&T efficiency improves over time, what are the differences between each transformation stage, and which will then move us one step further to making clear what matters for university S&T efficiency improvement. We attempt to provide a preliminary regulation summary of S&T efficiency improvement in different transformation stages.

## 3 Methodology

### 3.1 Super efficiency network SBM model

Existing studies analyzed the S&T transformation comparing research methods, and assessing the transformation ability and DEA efficiency evaluation [28, 29]. The traditional DEA model normally ignores the different stages of the whole transformation process and cannot measure the overall efficiency of the chain process. The efficiency evaluation of multi-department decision-making units is carried out separately, without considering the relationship among stages. This study uses the network SBM model [30] to deal with the efficiency of sub-units of inter-relationship in DMU (Decision Making Unit), which can effectively solve the problem of intermediate investments. Tone (2014) [31] proposed the non-radial and non-angle network SBM mode, whose advantage is that the efficiency value decreases monotonously with the change of input and output relaxation [32]. The synthetic efficiency of SBM model can be expressed by following equation:

$$min\rho_0^* = \frac{\sum_{k=1}^{K} w^k \left[1 - \frac{1}{m_k} \left(\sum_{i=1}^{m_k} \frac{s_{io}^{k-}}{x_{io}^k}\right)\right]}{\sum_{k=1}^{K} w^k \left[1 + \frac{1}{r_k} \left(\sum_{i=1}^{r_k} \frac{s_{ro}^{k+}}{y_{ro}^k}\right)\right]} \tag{1}$$

$$\text{s.t.} \begin{cases} \sum_{k=1}^{K} w^k = 1, w^k \geqslant 0 (\forall k) \\ x_0^k = X^k \lambda^k + s_0^{k-} & (k = 1, \cdots, K) \\ y_0^k = Y^k \lambda^k - s_0^{k+} & (k = 1, \cdots, K) \\ e\lambda^k = 1 (k = 1, \cdots, K) \\ S_0^{k+}, S_0^{k-}, \lambda^k \geqslant 0 & (\forall k) \\ z_0^{(k,h)} = Z^{(k,h)} \lambda^h & (\forall (k, h)) \\ z_0^{(k,h)} = Z^{(k,h)} \lambda^k & (\forall (k, h)) \end{cases}$$

$\rho_0^*$ is the overall efficiency of $DMU_0$; $X_{i0}^k \in R_+^{m_k}$ is the input vector of $DMU_0$ department K; $m_k$ is the input type of sector K; $Y_{r0}^k \in R_+^{r_k}$ is the output vector of sector K; $r_k$ is the output type of sector k; $s_{i0}^{k-}$ and $S_{r0}^{k+}$ are respectively relaxation variables of input and output; $w^k$ is the weight of department k; $\lambda^k \in R_n^+$ is a nonnegative vector; e is a constant, which means non Archimedean infinitesimal; $Z_0^{(k,h)}$ is the intermediate variable, which is the output of k sector and the input of h sector [2]. While the model cannot distinguish difference between effective DMU. In super-SBM [30] differences between valid DMU can be distinguished. The performance of DMU in super efficiency network SBM can be expressed by Eq (2). Among which the $S_{i0}^{k-*}$ and $S_{r0}^{k+*}$ are optimal input relaxation variables and optimal output relaxation variables respectively.

If $\rho_0^* = 1$, DMU$_0$ is effective, and the performance of transformation reaches the highest level; If $\rho_k = 1$, it shows that the K department in the DMU is efficient [2].

$$\rho_k = \frac{1 + \frac{1}{m_k}\left(\sum_{i=1}^{m_k} \frac{s_{io}^{k-*}}{x_{io}^k}\right)}{1 - \frac{1}{r_k}\left(\sum_{r=1}^{r_k} \frac{s_{ro}^{k+*}}{y_{ro}^k}\right)} (k = 1, \cdots, K) \qquad (2)$$

While the SBM network model cannot distinguish difference between effective DMU. In super-network SBM [30] differences between valid DMU can be distinguished. When P$_0$>1, the DMU is effective, the university scientific and technologycal transformation efficiency is the highest; If P$_k$>1, it indicates that the k-stage is effective. The super efficiency network SBM model in this paper evaluates the three stages efficiency of scientific and technologycal transformation efficiency in Chinese universities. This can help to find out the problems in different university scientific and technologycal transformation stages, and more accurate to find ways to improve efficiency.

From the efficiency index aspect, the input-output index of existing research usually considers R&D personnel, R&D funding, scientific research projects, academic papers, academic books, and so on, as input indicators of different efficiency calculation stages. The numbers of granted patents and sold patents are taken as output indicators [24, 33]. In this paper, the three-stage efficiency indicators are selected based on the existing research. According to the characteristics of different transformation stages, and considering the relationship between stages, different input-output indicators are considered (Table 1). The research period considered in this paper is 2008–2018, and the selected indicators cover all universities in the Yangtze River Economic Belt in China. Moreover, input-output data on the universities S&T transformation are taken from the "Compilation of scientific and technical statistics of colleges and universities (2009–2019) [34]"; data regarding the influencing factors are extracted from the "China City Statistical Yearbook (2009–2019) [35]" and the "Statistical Yearbook of China Science and Technology (2009–2019)" [36].

1. In basic knowledge acquisition stage, the main purpose of universities is basic research and knowledge creation. The input index are basic R&D expenditure and R&D personnel, the output of basic research R&D practitioners is mainly knowledge, and the output are papers and publications.

2. In applied technology innovation stage, universities apply and experiment new knowledge to test the feasibility of knowledge and whether it can solve practical problems. The input index is apply R&D personnel, R&D funding for apply and test Research, and the new created publications and papers. This stage mainly produces patents which can be the input of the next stage.

3. In Technology marketization stage, university uses the existing knowledge and application technology to serve the market and society demand, so the technology of the first two stages is

**Table 1. Indicator of S&T transform efficiency.**

| Phase | Input | Output |
|---|---|---|
| Phase 1 | Basic R&D Personnel; R&D Funding for Basic Research; | Research Publications; Research Papers; |
| Phase 2 | Apply R&D Personnel; R&D Funding for Apply and Test Research; Research Publications and Papers; | No. of Applied Patent; No. of Authorized Patent; No. of Awards; |
| Phase 3 | Application and Technology Service Personnel; Application and Technology Service Funding; No. of Authorized Patent; | Patent Licensing Income; Technology Transfer Income; Enterprise Investment Income; |

transformed into the market value. The input index are application and technology service personnel and funding, authorized patent. he main output of this stage is market income, including patent licensing income; technology transfer income and enterprise investment income.

## 3.2 Panel regression

The influencing factors of the university S&T achievements transformation involve complex aspects. Based on existing research and considering the reality of Chinese universities and data availability, the transformation efficiencies of three stages are selected as dependent variables, including the regional economic foundation, openness, industrial structure, quality of scientific research personnel, scientific research projects, and university international cooperation as the influencing factors. The panel regression model of the influencing factors on the university S&T achievements transformation progress is as follows

$$TE_{it} = \alpha + \beta_1 GDP_{it} + \beta_2 IND_{it} + \beta_3 OPEN_{it} + \beta_4 TAL_{it} + \beta_5 RES_{it} + \beta_6 INT_{it} + \varepsilon_{it}$$

$TE_{it}$ is the technology transformation efficiency of the $I_{th}$ city in the Yangtze River Economic Belt in the T year; $\alpha$ is a constant term; is the regression coefficient of each explanatory variable; $GDP_{it}$, $IND_{it}$, $OPEN_{it}$, $TAL_{it}$, $RES_{it}$, $INT_{it}$ represent GDP, industrial structure, openness, human resources, scientific research projects, and international cooperation of the $I_{th}$ city in the T year; $\varepsilon_{it}$ is a random disturbance.

1. $GDP_{it}$: The regional economic level shows a certain correlation with S&T transfer. The GDP is taken as the measurement index to test the influence of regional economic development on the transformation efficiency of university S&T.

2. $IND_{it}$: There may exist a certain correlation between industrial structure and the transformation of S&T. Based on existing research, this paper uses the proportion of tertiary industry to measure the optimization degree of industrial structure.

3. $OPEN_{it}$: Regional universities usually guarantee their competitiveness in the region by local resources investment. After entering the international market, universities must also actively transform technological achievements and carry out more efficient innovation activities to meet the demands of international competition. In this paper, FDI (Foreign Direct Investment) is used as an important indicator of regional international openness.

4. $TAL_{it}$: The quality of R&D talents is closely related to S&T transformation. In this paper, the proportion of R&D personnel with senior titles is used to measure the quality of R&D talents.

5. $RES_{it}$: The number of scientific projects could reflect the scientific research ability of universities. Therefore, this number is selected as an index to measure the university S&T transformation efficiency.

6. $INT_{it}$: International cooperation between universities is important to developing countries to absorb new knowledge and technology. The number of international exchanges and cooperation in universities is used to measure university international cooperation.

## 4 Overall efficiency of the Yangtze River Economic Belt

### 4.1 Feature of the case region

The Yangtze River Economic Belt includes 11 provinces and cities: Shanghai, Jiangsu, Zhejiang, Anhui, Jiangxi, Hubei, Hunan, Chongqing, Sichuan, Yunnan, and Guizhou. The Yangtze River Economic Belt accounts for 21% and 43% of China's land area and population, and it

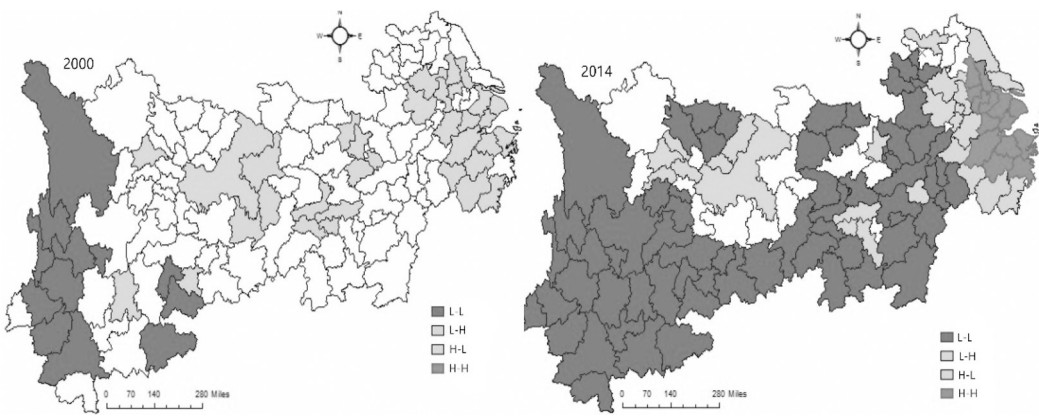

**Fig 2. Spatial co-relation of innovation in Yangtze Economic Belt.** Data source: Zou Lin, Zhu Yiwen. Research on Innovation Performance in Heterogeneous Region: Evidence from Yangtze Economic Belt in China [J]. Complexity, 2020 (2):1–9.

is a representative region of innovation in China. Comparing the R&D proportion of the Yangtze River Economic Belt with that of the entire country, the proportion of R&D expenditure in the Yangtze River Economic Belt accounts for 46% of China's expenditure, and the R&D personnel accounts for 45%.

The innovation output is considered an important indicator of the level of regional innovation, and shows a spatial distinction. Specifically, a trend-surface analysis of the innovation output from the northeast to the southwest of the Yangtze River Economic Belt reveals changes from high to low and a "core-periphery" feature. The Spatial Correlation Analysis of the Innovation Output shows that Moran's I index of the Yangtze River Economic Belt is all positive, and spatial dependence characteristics of the innovation output are becoming obvious. According to Lisa (Local indicators of spatial association) analysis, the high-value output agglomeration of the Yangtze River Economic Belt is not significant; no city is located in the High-High area, and Shanghai, Suzhou, Ningbo, Hangzhou, Wuhan, Changsha, and Chongqing show higher value characteristics than surrounding cities. The low-low cluster is prominently located in Lijiang, Linyi, Baoshan, and Lishui in Region 3. The high-value cluster of the innovative output continues to expand northward in Region 1, whereas the level of innovation output between Region 1 and Region 2 tends to be similar, but the level of agglomeration is relatively low (Fig 2). Otherwise, from the inter-regional innovation connection aspect, the centrality of the innovative network is enhanced, the network density decreased owing to the expansion of the network scale, the core of network structure is significant; network core-peripheral structure is increasingly significant.

Given the huge inner-regional heterogeneity, differences in innovation volume among regions are inevitable. Therefore, when analyzing areas with large spans and significant internal differences, we should not only focus on the absolute amount of resources but also consider whether knowledge or resources are reasonably configured. Therefore, when analyzing the S&T efficiency of universities and the influencing factors, this paper considers regional heterogeneity, analyzing the overall efficiency and evolution characteristics of the heterogeneous region.

## 4.2 Computation of overall S&T transform efficiency

This paper uses the super-efficiency network SBM model to calculate the efficiency of basic research achievements, the application of technology innovation, and the technology

marketization of university S&T achievements transformation of the Yangtze River Economic Belt from 2008 to 2018. Based on the characteristics of regional heterogeneity, this paper divides the Yangtze River Economic Belt into three regions: the Yangtze River Delta region, the central region, and the western region.

During 2008–2018, the average efficiency of the Yangtze River Delta (1.377) has been higher than that of the central (1.052) and western (1.35) regions (Table 2). However, the whole efficiency of the Yangtze River Economic Belt is not high, which is related to the organizational characteristics of universities. Given the openness of basic knowledge in universities and the fact that research application funding is not directly invested in the market, the proportion of funding for marketization is low. From the evolution perspective, the efficiency of the Yangtze River Economic Belt fluctuates and shows a slow downward trend, which shows that in recent years, university S&T achievements have not been effectively applied and marketized. Therefore, it is necessary to analyze the efficiency and its influencing factors in different transformation stages to improve the university S&T achievements transformation efficiency.

From 2008 to 2010, Zhejiang has become the region with the highest transformation efficiency in the Yangtze River Delta. The efficiency of the university S&T achievements transformation in Zhejiang has gradually increased (1.78 to 2.15). Since 2010, the overall efficiency in Anhui has been ahead of other regions in the Yangtze River Delta. By 2013, the efficiency has reached 3.29, far higher than the average level, and has become the leading region in the Yangtze River Delta. As an economic and financial center, Shanghai has outstanding advantages in scientific research and technology innovation and relatively high investment in scientific personnel and funding. However, there are still problems and limitations in university S&T achievements transformation. The efficiency of Shanghai changed from 1.187 to 1.007. Therefore, Shanghai must solve the problem of insufficient transformation and utilization of S&T achievements to realize the high-level development of regional innovation. In central regions, Hubei led other regions in the transformation rate of scientific research achievements (2.69) in 2008 and then decreased. Recently, Hunan showed an upward trend, reaching 1.9 in 2014, which is higher than the average level of central regions. The overall efficiency of the western region is higher than that of the central regions, and the universities in Yunnan and Guizhou have higher efficiency. This is consistent with the conclusion that the main goal of inter-

**Table 2. Efficiency of university S&T achievements transformation in Yangtze River Economic Belt.**

| Region | 2008 | 2009 | 2010 | 2011 | 2012 | 2013 | 2014 | 2015 | 2016 | 2017 | 2018 | average |
|---|---|---|---|---|---|---|---|---|---|---|---|---|
| Shanghai | 1.1879 | 1.4035 | 1.0461 | 1.1531 | 1.1428 | 1.1585 | 1.1218 | 1.0651 | 1.0644 | 1.0346 | 1.0070 | 1.126 |
| Jiangsu | 1.0284 | 1.1174 | 1.8096 | 1.2178 | 1.2829 | 1.2966 | 1.3452 | 1.5308 | 1.1869 | 1.1098 | 1.2056 | 1.285 |
| Zhejiang | 1.7867 | 1.2855 | 2.1547 | 1.5099 | 1.2743 | 1.3938 | 1.4393 | 1.2080 | 1.3311 | 1.2956 | 1.1682 | 1.441 |
| Anhui | 1.3560 | 1.1408 | 1.0471 | 1.0148 | 1.6595 | 3.2958 | 2.2962 | 1.8320 | 2.2988 | 1.1787 | 1.1015 | 1.656 |
| Delta average | 1.3398 | 1.2368 | 1.5144 | 1.2239 | 1.3399 | 1.7862 | 1.5506 | 1.4090 | 1.4703 | 1.1547 | 1.1206 | 1.377 |
| Jiangxi | 1.2292 | 1.1503 | 0.5791 | 1.0685 | 1.0764 | 1.0488 | 0.5416 | 0.4766 | 0.7116 | 0.8795 | 1.2568 | 0.911 |
| Hubei | 2.6915 | 1.0267 | 1.0040 | 1.0156 | 0.8839 | 1.1781 | 0.8222 | 0.7424 | 1.0259 | 1.1179 | 1.0421 | 1.141 |
| Hunan | 0.7296 | 1.0306 | 1.0082 | 1.3706 | 1.0828 | 0.7695 | 1.9093 | 1.0050 | 1.0606 | 1.1086 | 1.0627 | 1.103 |
| Middle average | 1.5501 | 1.0692 | 0.8638 | 1.1516 | 1.0144 | 0.9988 | 1.0910 | 0.7413 | 0.9327 | 1.0353 | 1.1205 | 1.052 |
| Chongqing | 1.0236 | 1.0732 | 1.0509 | 1.7904 | 1.2338 | 1.0042 | 0.8294 | 0.7196 | 1.2008 | 1.6505 | 1.0515 | 1.148 |
| Sichuan | 1.1105 | 1.6748 | 1.4765 | 1.3177 | 1.3713 | 1.1080 | 1.1585 | 1.2147 | 1.1217 | 1.1020 | 1.0676 | 1.248 |
| Guizhou | 1.3041 | 1.2076 | 1.2547 | 1.4275 | 1.5989 | 1.0403 | 2.1277 | 2.2233 | 2.8951 | 2.3688 | 1.2058 | 1.696 |
| Yunnan | 1.2926 | 1.7601 | 1.4837 | 1.8999 | 1.4709 | 1.1238 | 1.1316 | 1.0283 | 1.1237 | 1.0703 | 1.0510 | 1.312 |
| West average | 1.1827 | 1.4289 | 1.3164 | 1.6089 | 1.4187 | 1.0691 | 1.3118 | 1.2965 | 1.5853 | 1.5479 | 1.0940 | 1.351 |

regional network innovation cooperation in western regions is to realize the new technology application and marketization (Lin, 2016).

Regions with relatively insufficient economic basis and innovation input are not deficient in the university S&T achievements transformation efficiency, and regions with rapid economic development and higher scientific investment do not necessarily achieve high efficiency. Therefore, we need to think about which stages in the process affect the efficiency improvement of the regions, considering the three main stages of basic knowledge transformation, applied technology innovation, and technology marketization. Based on the analysis of the trend of the university S&T achievements transformation efficiency in the Yangtze River Economic Belt, this paper further attempts to analyze the stage efficiency and its main influencing factors. This paper tries to find out the stage problems in the university S&T achievements transformation process in the Yangtze River Economic Belt and puts forward some possible suggestions to improve the efficiency.

## 5 Stage analysis of university S&T transformation efficiency

### 5.1 Computation of three-stage S&T transform efficiency

From the perspective of the computation of the efficiency of the three stages of university S&T achievements transformation, the average efficiency level of basic knowledge transformation decreased in the Yangtze River Economic Belt (Fig 3). The average efficiency in the Yangtze River Delta (1.288) is higher than in the central (1.084) and the western regions (1.146). From the evolution perspective, Zhejiang and Shanghai in the Yangtze River Delta have had obvious advantages from 2008 to 2012. After 2012, Anhui and Jiangsu also shared some advantages. It proves that the basic research input promotes the improvement of basic knowledge and realizes an efficient transformation of basic scientific research. However, Jiangxi (0.888) in the central regions, Chongqing (1.089), and Guizhou (1.099) in the western regions have lower values than the regional average level. Therefore, how to improve the efficiency of basic knowledge transformation in these areas is the key to improve university S&T achievements transformation in the Yangtze River Economic Belt.

There are significant differences in the efficiency of technological innovation among different regions of the Yangtze River Economic Belt (Fig 4). The efficiency of technological innovation in the central regions is insufficient (0.993), which is far lower than that in the Yangtze River Delta (1.514) and the western regions (1.282). Therefore, how to improve the efficiency of university technological innovation in the central regions, and realize the efficient utilization of applied research resources, such as applied research funding, personnel, and papers are

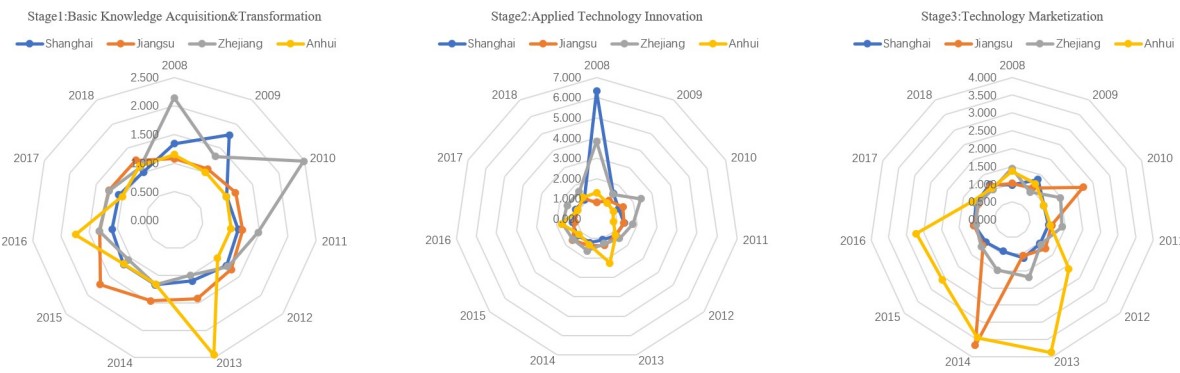

**Fig 3. Efficiency of university basic knowledge transformation stage in Yangtze River Economic Belt.**

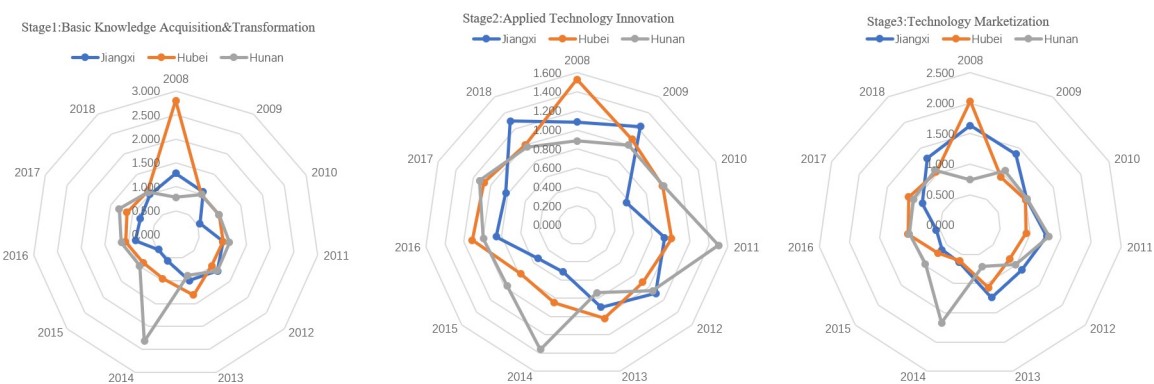

**Fig 4. Efficiency of university applied technology innovation stage in Yangtze River Economic Belt.**

the key to improve the overall efficiency of university S&T achievements transformation in the Yangtze River Economic Belt.

Whether the applied technology can be successfully marketized and produce value for regions and organizations is an important manifestation of the success of university S&T achievements transformation. In most areas of the Yangtze River Economic Belt, the efficiency of technology marketization has increased (Fig 5); the average efficiency of the Yangtze River Delta, the central regions, and the western regions were 1.428, 1.047, and 1.435, respectively. Among them, Anhui (1.968) and Guizhou (1.983) have higher efficiency of technology marketization, which indicates that the new patents generated by the application of the technology innovation stage can realize marketization efficiently through experimental research and, then, obtain better economic benefits. However, the marketization efficiency in Shanghai, Hubei, and Chongqing is insufficient, and it is lower than the regional average level. Therefore, improving the technology marketization of these regions is the key to realize the high-efficiency technology marketization of the Yangtze River Economic Belt.

### 5.2 Analysis of the influencing factors of stage efficiency

According to the panel regression analysis, the regional economic level shows a strong positive significance in stage one (Table 3), which has a positive effect on basic knowledge transformation. This conclusion is consistent with the efficiency analysis of regional scientific knowledge transformation. A better foundation of regional macro-economic development is conducive to

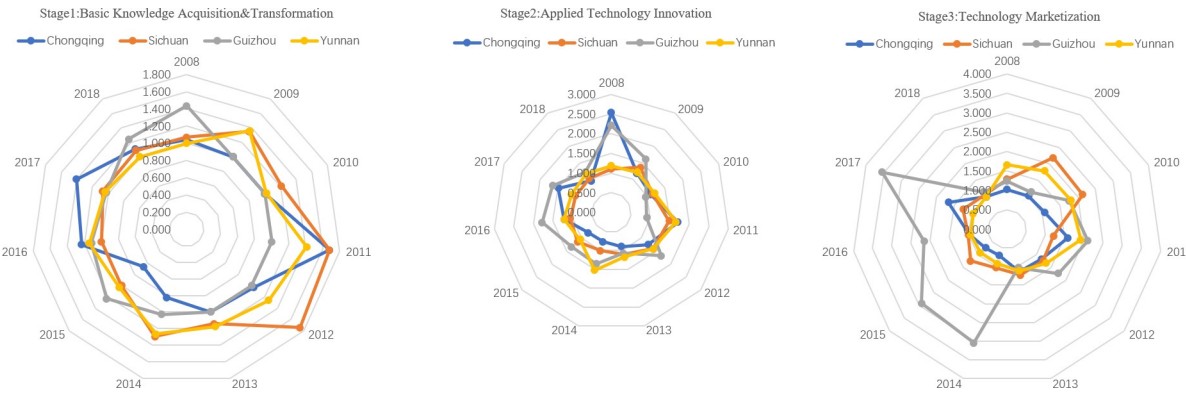

**Fig 5. Efficiency of university technology marketization stage in Yangtze River Economic Belt.**

**Table 3. Influence factor of university S&T transformation efficiency.**

|  | Phase 1 | | Phase 2 | | Phase 3 | |
|---|---|---|---|---|---|---|
|  | **Coef.** | **P>\|t\|** | **Coef.** | **P>\|t\|** | **Coef.** | **P>\|t\|** |
| GDP | 0.703** (0.322) | 0.031 | -0.67 (0.550) | 0.228 | -0.000 (0.000) | 0.956 |
| Industry Structure | -1.481 (1.144) | 0.198 | -4.65** (1.955) | 0.019 | -1.593 (1.574) | 0.312 |
| FDI | 0.353* (0.180) | 0.053 | 0.65** (0.308) | 0.038 | 0.000 (0.000) | 0.838 |
| Talents | 1.334 (1.712) | 0.438 | 6.14** (2.926) | 0.038 | 4.143* (2.401) | 0.084 |
| Research Project | -0.240 (0.165) | 0.148 | -0.37 (0.281) | 0.187 | -0.000 (0.000) | 0.401 |
| International Cooperation | 0.205 (0.188) | 0.278 | 0.15 (0.321) | 0.645 | 0.000 (0.000) | 0.387 |
| Cons | 3.325** (1.603) | 0.041 | 3.87 (2.740) | 0.160 | 0.644 (0.975) | 0.509 |
| Hausman P | 0.0015 | | 0.000 | | 0.658 | |
| Model | Fixed efficiency | | Fixed efficiency | | Random efficiency | |

Note

* means P<0.1

** means P<0.05.

the basic knowledge transformation of universities, whereas a relatively insufficient economic basis restricts the transformation of basic research. This is one of the reasons why the efficiency of basic scientific research transformation in the Yangtze River Delta is higher than that in the central and western regions. However, the role of the regional economic basis in the technology application and marketization stages is not significant. It shows that the overall transformation efficiency of universities cannot focus only on improving economic input.

Regional openness has a positive effect on the transformation of university basic knowledge and technological innovation (Table 3). With the improvement of openness, universities can have more opportunities to participate in international industry-university-research cooperation and realize international knowledge flow and technology innovation. The improvement of international relations also requires universities to continuously improve the basic knowledge transformation efficiency to better meet the requirements of participating in international knowledge competition.

The areas where the transformation efficiency of basic research is insufficient, such as Jiangxi in the central region, Chongqing and Guizhou in the western region, it is difficult to improve the regional economic base in the short term. Therefore, improving regional openness is a more effective way. The region can introduce transnational investment and take the technology demand of international enterprises as the guidance, which in turn promotes the research and transformation of basic knowledge in colleges and universities. The central region of the Yangtze River has the problem of difficult technological innovation. The promotion of regional openness also has a significant impact on the improvement of application technology innovation efficiency. Therefore, improving regional openness is the key to solve the application technology innovation of universities in the central region.

The quality of R&D talents has a significant positive effect on the application of technological innovation and technology marketization (Table 3). Paying attention to the quality of R&D talents is key to improve the efficiency of regional technological innovation and marketization. Human resources are essential to ensure the productivity of university scientific research; however, the increase of the input of human resources alone cannot improve the efficiency of innovation and, sometimes, it will cause redundancy or waste. Other factors to be considered include a higher level of scientific R&D talents, a higher quality of scientific expenditure, and a higher efficiency of university technological innovation and marketization. The Yangtze River

Delta is China's talent pool, which gathers high-end technical talents from China and abroad, which is the basis for its successful marketization of applied technology. The purpose of basic research and technological innovation in western region is to realize the marketization of technology and create market value [37]. The western development and support policies of the Chinese government have also provided a large number of high-quality scientific research talents for the western region, which has improved the quantity and quality of talents in the western region, and successfully realized the marketization of technological achievements in the western region. Therefore, regions with insufficient efficiency of technological innovation or marketization should pay more attention to the quality of R&D talents and ensure that the talents in universities have a frontier theoretical knowledge system.

Some studies on the relationship between the structure of the tertiary industry and the transformation of S&T indicate that the proportion of the tertiary industry is positively correlated with university S&T transformation [38]. The Yangtze River Economic Belt has formed manufacturing industrial clusters of electronic information, high-tech equipment, automobile, textile, and clothing. The development of the high-tech industry and equipment manufacturing industry is the basis of knowledge innovation, technology application, and transformation. Therefore, the special industry structure of the Yangtze River Economic Belt leads to different performances of its influencing factors. Therefore, the structure of the tertiary industry, in this case, is not positively related to university S&T transformation. This research result is based on the development foundation of the high-tech industry in the Yangtze River Economic Belt and is consistent with the fact that the manufacturing industry plays a guiding role in the region.

International cooperation and scientific research projects do not show a strong influence on the transformation of scientific research achievements of regional universities. The international cooperation of universities shows a positive correlation. Scientific research needs to rely on international platforms. International communication and cooperation have positive effects on scientific research [39]. However, the proportion of Chinese participants in international exchanges in universities is relatively low. Therefore, the impact on university S&T transformation efficiency is still insufficient, which is also one of the key points that Chinese universities need to consider in order to improve the S&T transformation efficiency. The number of research projects is negatively correlated with efficiency and it is not significant, which shows that the blind increase in the number of research projects cannot achieve efficient transformation. It may be more effective to think about how to improve the quality of projects or properly allocate projects in different regions with high efficiency.

## 6 Discussion and conclusion

This paper proposed a three-stage efficiency analytical framework for regional university S&T transformation, which regards the transformation of university S&T as a complex process with mutual relations. The whole process is divided into three main stages: basic knowledge acquisition and transformation, the application of technology innovation, and technology marketization. The super-efficiency network SBM model is used to estimate the efficiency of university S&T transformation in the Yangtze River Economic Belt. It is helpful to accurately measure both the stage efficiency and the whole process in such a heterogeneous region in China.

The university S&T transformation in the Yangtze River Economic Belt is not high, and in recent years, the efficiency has not been effectively improved. Although this is related to the characteristics of university knowledge openness, it must be acknowledged that the scientific research resources invested in universities in the Yangtze River Economic Belt are not efficiently utilized. There are regional efficiency differences within the Yangtze River Economic Belt; the efficiency of the Yangtze River Delta is slightly higher than that of the central and

western regions, and the transformation efficiency of the central regions is the lowest. Shanghai is the financial center of China, with outstanding economic and scientific research advantages. How to realize high transformation efficiently and avoid waste or redundancy of resources is the main problem.

There is a certain correlation between the economic foundation and overall transformation efficiency; however, an insufficient economic foundation does not necessarily lead to low efficiency. The transformation of university S&T should not only focus on whether there are funds or personnel input. To effectively promote university S&T transformation and avoid waste of scientific resources, targeted adjustments should be made according to regional heterogeneity and transformation stage.

In the stage of basic knowledge transformation, the efficiency of the Yangtze River Delta is higher than the central and western regions, and the efficiency of transformation in Anhui and Jiangsu is constantly improving. The advantages provided by the regional economic foundation in the Yangtze River Delta play an important role. In addition, the advantages determined by regional openness make the basic knowledge of universities in the Yangtze River Delta realize high-efficiency transformation. Although the economic foundation of the central and western regions does not offer a comparative advantage, universities can still improve the basic knowledge transformation efficiency by strengthening inter-regional, especially international knowledge cooperation and knowledge flow. The allocation of university scientific resources in China has not yet reached the optimal level, and still needs to be further optimized.

The application of technological innovation and successful marketization are crucial to reflect the value of scientific achievements in universities. The efficiency of the marketization of scientific achievements of the universities in the Yangtze River Economic Belt has risen slowly, which shows that universities in China gradually realize the importance of combining theory with practice and creating market value. The efficiency of marketization in western regions is the highest, which is consistent with the purpose of technological innovation. The international openness of the Yangtze River Delta still makes it gain advantages in the application of technological innovation. At the same time, the regional openness makes it absorb a large number of high-quality inter-regional or international talents. The flow of talents brings more knowledge and technology flow, and further improves the efficiency of the regional technology application of universities.

To achieve a highly efficient transformation of university S&T, Chinese universities should not only focus on the amount of scientific investment. Most universities produce rich scientific outputs, but these outputs have not been successfully transformed owing to their low quality. This requires universities to pay more attention to the source of knowledge production and avoid that the quantity of basic scientific achievements is large, while it is difficult to transform it because of insufficient research quality.

Based on basic research, it is necessary to enhance the innovation ability of applied technology in universities, produce high-level scientific achievements, and strengthen the applicability and transformation of basic knowledge. Finally, creating real market value for the regional economy is essential for Chinese universities in the future. Innovation and knowledge creation are critical in triggering fundamental economic and social changes [40]. Since Etzkowitz [41] put forward the triple helix theory of organizational innovation in the 1990s, studies have been focusing on how to realize knowledge production and transformation efficiently. Universities are important members of the innovation system in a knowledge-based economy, and regions are considered important space carriers for innovation activities. The regional innovation system is composed of enterprises, universities, and research institutions, which can continuously produce innovation [42]. Knowledge creation, absorption, and transformation of research institutions, universities, technology transfer institutions, enterprises, and other organizations

in the regional innovation system are conducive to the improvement of innovation efficiency in the region. Therefore, the analysis and exploration of the mechanism of university knowledge production and transformation will help to understand and improve regional innovation.

The realization of university S&T transformation is a complex process. Due to data availability, there are still some limitations in the selection of indicators in the analysis of the influencing mechanisms. Moreover, the analysis of spatial heterogeneity only compares differences in three transformation stages but does not consider internal spatial interactions and network relationships. These are topics for further research.

## Supporting information

**S1 Data.**
(XLSX)

## Author Contributions

**Writing – original draft:** Lin Zou.

**Writing – review & editing:** Yi-Wen Zhu.

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
