## [Decision Letter · Decision Letter 0]

15 Jun 2021

PONE-D-21-13248

Universities’ Scientific and Technological Transformation in China: Its Efficiency and Influencing Factors in the Yangtze River Economic Belt

PLOS ONE

Dear Dr. ZOU,

Thank you for submitting your manuscript to PLOS ONE. After careful consideration, we feel that it has merit but does not fully meet PLOS ONE’s publication criteria as it currently stands. Therefore, we invite you to submit a revised version of the manuscript that addresses the points raised during the review process.

We look forward to receiving your revised manuscript.

Kind regards,

Ming Zhang, Ph.D.

Academic Editor

PLOS ONE

Journal Requirements:

Reviewers' comments:

Reviewer's Responses to Questions

**Comments to the Author**

1. Is the manuscript technically sound, and do the data support the conclusions?

Reviewer #1: Partly

2. Has the statistical analysis been performed appropriately and rigorously? 

Reviewer #1: No

3. Have the authors made all data underlying the findings in their manuscript fully available?

Reviewer #1: Yes

4. Is the manuscript presented in an intelligible fashion and written in standard English?

Reviewer #1: No

5. Review Comments to the Author

Reviewer #1: Using three-stage network SBM model, this paper analyzes the efficiency of university S&T transformation in the Yangtze River Economic Belt. Although the author(s) did some interesting analysis, the contribution of this paper is not clear. I hope the author(s) find the following comments and suggestions helpful.

1. In the abstract, the authors write "Avoiding the problem of considering the input and output of university S&T transformation as a "black box" and neglecting the links among different transformation stages.". However, it is well known that DEA methods (including SBM) have the disadvantage of considering the production process as a black box. I would like the authors to explain this conclusion.

2. Please describe in detail the embedding position of this paper in the existing studies and the marginal contribution made by this paper.

3. The introduction of the Yangtze River Economic Belt in section 4.1 is better moved to the background section.

4. What are the standards for dividing the central and western regions in the paper? Why are there only 3 and 4 provinces in the central and western regions respectively? In addition, the result that efficiency is higher in the western region than that in the central region is inconsistent with experience, please provide an explanation.

5. Please elaborate on the criteria for the selection of explanatory variables in panel regressions.

6. Why do almost all factors have no significant effect on the efficiency of the third phase?

7. Regression results after adding fixed effects should be reported in table 3.

6. PLOS authors have the option to publish the peer review history of their article (what does this mean?). If published, this will include your full peer review and any attached files.

Reviewer #1: No

---

## [Author Response · Author response to Decision Letter 0]

1 Oct 2021

Thank you very much for the comments of the reviewer, which help us rethinking our research deeper. The suggestions for this manuscript help a lot to make the article step further.

best regards

---

## [Decision Letter · Decision Letter 1]

1 Dec 2021

Universities’ Scientific and Technological Transformation in China: Its Efficiency and Influencing Factors in the Yangtze River Economic Belt

PONE-D-21-13248R1

Dear Dr. ZOU,

We’re pleased to inform you that your manuscript has been judged scientifically suitable for publication and will be formally accepted for publication once it meets all outstanding technical requirements.

Kind regards,

Ming Zhang, Ph.D.

Academic Editor

PLOS ONE

Additional Editor Comments (optional):

Reviewers' comments:

Reviewer's Responses to Questions

**Comments to the Author**

1. If the authors have adequately addressed your comments raised in a previous round of review and you feel that this manuscript is now acceptable for publication, you may indicate that here to bypass the “Comments to the Author” section, enter your conflict of interest statement in the “Confidential to Editor” section, and submit your "Accept" recommendation.

Reviewer #1: (No Response)

2. Is the manuscript technically sound, and do the data support the conclusions?

Reviewer #1: (No Response)

3. Has the statistical analysis been performed appropriately and rigorously? 

Reviewer #1: (No Response)

4. Have the authors made all data underlying the findings in their manuscript fully available?

Reviewer #1: (No Response)

5. Is the manuscript presented in an intelligible fashion and written in standard English?

Reviewer #1: (No Response)

6. Review Comments to the Author

Reviewer #1: (No Response)

7. PLOS authors have the option to publish the peer review history of their article (what does this mean?). If published, this will include your full peer review and any attached files.

Reviewer #1: No

---

## [Editor Report · Acceptance letter]

6 Dec 2021

PONE-D-21-13248R1 

Universities’ Scientific and Technological Transformation in China: Its Efficiency and Influencing Factors in the Yangtze River Economic Belt 

Dear Dr. ZOU:

I'm pleased to inform you that your manuscript has been deemed suitable for publication in PLOS ONE. Congratulations! Your manuscript is now with our production department. 

Kind regards, 

on behalf of

Dr. Ming Zhang 

Academic Editor

PLOS ONE